# A Blockchain Storage Architecture Based on Information-Centric Networking

Hangwei Feng [1,2], Jinlin Wang [1,2] and Yang Li [1,2,*]

1   National Network New Media Engineering Research Center, Institute of Acoustics, Chinese Academy of Sciences, No. 21, North Fourth Ring Road, Haidian District, Beijing 100190, China
2   School of Electronic, Electrical and Communication Engineering, University of Chinese Academy of Sciences, No. 19(A), Yuquan Road, Shijingshan District, Beijing 100049, China
*   Correspondence: liyang@dsp.ac.cn

**Abstract:** Blockchain technology is a unique distributed ledger technology that has been widely used in various areas. With the increase in data on the blockchain and the append-only nature of the blockchain, the traditional blockchain's full replica storage technique leads to blockchain storage scalability problem. Existing methods prioritize minimizing the storage strain on blockchain nodes while ignoring the availability of data, resulting in a lengthy average response time for users to access the blockchain. To address the shortcomings, this paper proposes an Information-Centric Networking-based blockchain storage architecture. The architecture uses the enhanced resolution system for community division to build blockchain node partitions and store blockchain ledgers in the underlying network. It introduces virtual chain for rapid blockchain indexing and adopts a collaborative block replica deletion algorithm across neighboring partitions, including replica number decision based on blockchain access decay characteristics and replica deletion based on resource relationship. Finally, we compare and analyze the proposed blockchain storage architecture with BC-store and KASARASA, and the results demonstrate that this architecture has significantly lower average access time than others. The replica data volume of this method is reduced by 57.2% compared to the full replica policy, but the access time is only 5.2% slower when compared to the full replica policy, which substantially increases the replica storage utilization.

**Keywords:** blockchain storage; information-centric networking; replica delete; decay

## 1. Introduction

Blockchain is a special distributed ledger technology that originated from the Bitcoin project invented by Satoshi Nakamoto. It connects blocks with hash pointers and creates a unique chain structure that can be shared and maintained by various parties [1]. Blockchain technology has been widely and actively used in a variety of areas, including: (1) cryptocurrency [2,3], (2) data management [4,5], and (3) information security [6].

However, as blockchain becomes more widely used in many settings, its weaknesses become increasingly obvious. Only two of the three technological paradoxes associated with blockchain, namely, decentralization, security, and scalability, can be resolved. This is known as the "blockchain impossible triangle" [7]. Earlier blockchain projects, such as Bitcoin and Ethereum, prioritize decentralization and security over scalability. This causes throughput, storage, and network scalability bottlenecks. The blockchain storage scalability problem has hampered blockchain technology development.

The primary cause of the storage scalability problem is that each blockchain node in the traditional blockchain system stores a complete ledger, which uses a full copy storage mode with high redundancy. This method dramatically improves the transparency of blockchain data and its accessibility. However, each blockchain node must synchronize the most recent blockchain ledger data. Using the public blockchain as an example, Bitcoin's

ledger data has reached close to 300 GB and Ethereum has surpassed 1TB by the year 2020. The data has increased and will continue to do so.

In addition, as blockchain technology evolves, efficient consensus algorithms with low consumption emerge, and the throughput bottleneck is continually being eliminated. This will dramatically accelerate the expansion of blockchain data. The storage scalability problem has restricted the blockchain's development on practical application and future direction [1,7].To conclude, the blockchain storage scalability problem consists of the following:

- **Single node storage bottleneck problem:** Every blockchain node must store the entire blockchain ledger. The amount of ledger data is excessive for a node.
- **Scalability of the blockchain system:** Blockchain node storage bottleneck and retrieval problems will restrict the new nodes from joining the blockchain system. It destroys the blockchain's decentralized characteristics and directly impacts its scalability.
- **Storage resource waste:** Thousands of blockchain nodes are required to store ledger data, resulting in huge storage resource waste.

Various academic teams have proposed solutions to blockchain storage scalability issues. These solutions can be categorized as off-chain or on-chain schemes according to how blockchain data is managed.

Off-chain schemes use existing storage systems to store blockchain ledger data, including blockchain storage solutions based on Distributed Hash Table (DHT), solutions based on Inter Planetary File System (IPFS), and cloud-based blockchain storage solutions. Zyskinds proposes to store blockchain data using DHT. Zheng QH, Ali MS, and Xu QQ use IPFS, a distributed storage system, to expand the blockchain storage capacity. Ali M introduces cloud storage to optimize blockchain storage. However, both cloud and IPFS solutions have limitations. Cloud lacks decentralization, and IPFS is difficult to meet the time requirement when storing blockchain data using .

On-chain schemes mainly use compression or sharding technologies to reduce the amount of stored data in the blockchain node, including collaborative storage solutions and light node solutions. Collaborative storage solutions include the blockchain storage architecture based on network coding–based distributed storage (NC-DS), the distributed blockchain storage strategy based on MCMC random algorithm, and the segmented blockchain storage architecture. The light node solutions include Bitcoin's light node and Ethereum's light node, and an archive node solution.

Current research focuses on reducing the data amount stored by blockchain nodes but does not consider the availability of blockchain data. Off-chain schemes and on-chain schemes both utilize the peer-to-peer (P2P) overlay network. It has various benefits, including storage capability, low cost, and robustness. However, it disregards the underlying physical address information when obtaining data. It is challenging for DHT to meet the needs for rapid data indexing, and IPFS has a static copy mechanism, which causes the stored data acquisition inefficiency.

With the ongoing growth of the network in recent years, information-centric networking (ICN) has emerged. ICN separates identity and locator, and possesses the hierarchical enhanced resolution system and in-network storage capabilities. Due to its autonomy, some ICN architectures have the ability to provide on-site data computation and storage with guaranteed latency and performance. These networks offer a novel solution to the blockchain storage scalability problem. This paper focuses on solving the blockchain storage scalability problem using ICN network, which may be applied to a variety of existing public blockchain storage scenarios. In addition, the network referred to in this research is the ICN network with enhanced resolution system. The contributions are as follows:

- To address the user access problem in blockchain storage architecture, this paper establishes a blockchain storage architecture using network storage capacity and enhanced resolution capability. It introduces virtual chain to index block to ensure the proximity of ledger data access.



- For blockchain replica management algorithms' poor access performance problem, this paper proposes a block replica number decision mechanism based on blockchain access decay relations. It introduces the forgetting model to design block decay factor and use decay factor for block replica number decision, dramatically reducing the computational overhead of replica estimation.
- For the block replica deletion problem, this paper establishes a replica deletion model based on the replica deletion loss and load state, and proposes a collaborative replica deletion algorithm with greediness to ensure system performance and load balancing.

The paper structure is as follows. Section 2 presents existing blockchain storage solutions and the ICN network. Section 3 presents the novel blockchain storage architecture, including the overall architecture, the community division mechanism based on the enhanced resolution system, the virtual chain structure, and the collaborative block replica deletion algorithm. In Section 4, the proposed blockchain storage architecture is analyzed experimentally. In Section 5, conclusions and future research are discussed.

## 2. Related Works

### 2.1. Blockchain Storage Architecture

For blockchain storage scalability problems, various research teams have proposed solutions, which can be separated into off-chain storage and on-chain storage, as shown in Table 1.

**Table 1.** Blockchain storage scalability problem solutions.

| Classification | Blockchain Storage Solutions | Reference |
|:---:|:---:|:---:|
| off-chain | DHT-based blockchain | [8,9] |
| | IPFS-based blockchain | [10–12] |
| | cloud-based blockchain | [13] |
| on-chain | collaborative storage | [14–17] |
| | light node | [2,3,18] |

Off-chain storage solutions use existing storage systems to store blockchain ledger data, including DHT-based, IPFS-based, and cloud-based blockchain storage solutions. Zyskind [8] modifies the traditional blockchain storage mode by separating the storing data and data reference, and designs an off-chain storage mode based on DHT. Abe [9] proposes a KARAKASA architecture, which organizes blockchain nodes with limited storage capacity into DHT clusters to to decrease the storage burden on single blockchain nodes. Zheng [10] and Chou [12] both introduce IPFS to solve the blockchain storage scalability problem. Ali [11] proposes a modular alliance architecture based on blockchain and IPFS to address the Internet of Things privacy issues. It eliminates the centralized management mode in IoT data and addresses the standard blockchain network deficiency.

Ali M [13] provides a cloud-based storage solution that stores the data itself in the cloud and the data hash value in the blockchain node. However, due to the centralization of public clouds, academics are paying more attention to DHT-based and IPFS-based solutions. However, both these use the underlying overlay network and DHT retrieval mechanism. They ignore the underlying physical address information, resulting obtaining low efficiency.

On-chain storage schemes mainly use compression or sharding technologies to reduce the stored data amount in a node, including collaborative storage solutions and light node solutions. Dai [14] proposes a blockchain storage architecture based on NC-DS. It uses network encode and store blocks in a distributed network. However, when obtaining transactions in network, this architecture will generate a large number of network requests, resulting in massive network pressure. Guo [15] proposes a blockchain storage system optimization scheme based on a redundant reminder system, which significantly reduces

blockchain node storage data amount and takes fault tolerance into account. Zhao [16] proposes a distributed blockchain storage technique based on the MCMC random algorithm, as well as a semi-full node. The random algorithm can delete part of blockchain ledger data, while the random algorithm ensures the load balance in the whole network. Jia [19] proposes a blockchain storage capacity scalability model, in which the blockchain ledger is fragmented and stored to different blockchain nodes, and the location is managed on the blockchain to solve the reliability problem.

Both Bitcoin and Ethereum have proposed light node schemes. Bitcoin [2] proposes a scheme combining light node and full node. The full node holds the entire blockchain ledger, whereas the light node just stores block header data and obtains full transaction data from the full node. Because of Ethereum's unique account model, Ethereum [3] proposes full node, light node, and archive node. The full node stores the complete account state and relevant transaction records, while the light node stores block headers and Merkel-tree related to transactions. The archive nodes store each block height state snapshots. However, these schemes do not fundamentally solve the block storage problem. Because more blockchain nodes will choose to become light nodes or join the mining pool, the full nodes number will drastically reduce [18]. It will directly lead to blockchain system centralization. This on-chain approach leads to blockchain nodes accessing blocks that cannot be accessed locally and need to be fetched remotely, resulting in high average response time and affecting blockchain data availability.

*2.2. ICN Network*

Due to the semantic overload of internet protocol (IP) addresses, the traditional network architecture has issues with scalability, mobility, and security, making it challenging to accommodate the growing demand from new application services.

In this context, ICN networks [20] emerges, whose core idea is to separate identity (ID) and locator (LOC). In the existing ICN architecture, name resolution methods are divided into two types based on name resolution and content routing: Name-Based Routing (NBR) and Standalone Name Resolution (SNR). The former employs hierarchical naming, where content requests are directed to content providers before being delivered to content requestors along the request's reverse channel, such as CCN and NDN [21]. The latter uses a flat naming approach to decouple name resolution and content routing, with ID resolution through the name resolution system and then routing through the LOC. This approach requires maintaining resolution system mapping ID and LOC, represented by DONA [22] and SEANet [23].

The name resolution system includes the global resolution system and enhanced resolution system. The global resolution system is responsible for the full amount of storage and is generally deployed in the cloud to ensure name resolution accessibility. DONA uses a global resolution system to manage identity and locators mappings. On this basis, some ICN networks have developed enhanced resolution systems [24] in order to meet more resolution requirements. NetInf combines global and enhanced resolution, using a multi-layer distributed hash table to query LOC. SEANet develops a local enhanced name resolution mechanism to provide a one-to-many relationship between ID and LOCs within the resolution domain. It achieves deterministic latency in limited domains by speeding up the name resolution process. The enhanced resolution system is shown in Figure 1. The bottom layer represents the entity network, and the above layers represent the different layers of the enhanced resolution system. The black dots represent the resolution nodes, the white dots represent the in-network devices, and the solid circles represent the resolution service areas at each layer.

The enhanced resolution system is locally oriented and organizes resolution nodes in a multi-layer tree structure, which can provide multi-level deterministic time-delayed enhanced resolution services, and the nodes at different levels provide different levels of resolution services. Users can obtain resolution services from multiple hierarchical resolution nodes according to requirements.

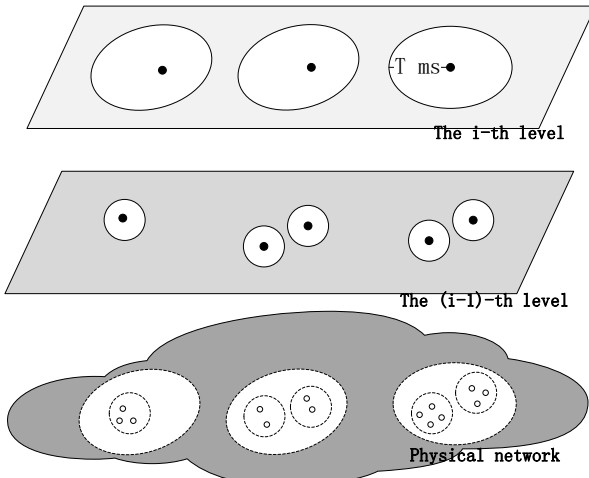

**Figure 1.** Enhanced name resolution system.

The resolution service level is reflected in the resolution latency. The higher the tier, the higher the maximum resolution latency that can be guaranteed and the larger the service area. Each resolution node provides resolution services only to the nodes in its service area. The resolution service area means that the resolution node can guarantee that the resolution latency is within the determined delay of the corresponding service when providing resolution service to any node within the area. The resolution service determination latency for the i-th level resolution node is T ms. when a network node in the service area performs query resolution service to this resolution node, the latency is less than or equal to T ms . At the same time, the enhanced resolution system service area is nested, and the service area of the upper level resolution service can include the lower level resolution service.

In addition, with the improvement of hardware capability, network devices gradually have computing and storage capabilities in addition to supporting routing and forwarding functions. These computing and storage resources distributed on forwarding devices can provide on-site computing and storage for networks that are self-organizing with resources that can quickly meet time-sensitive services.

## 3. Architecture Design

### 3.1. The Overall Architecture

To address the problem in the existing blockchain storage architecture, this paper proposes a blockchain storage architecture based on the ICN network, as shown in the following Figure 2.

The lowest layer is the ICN network, and this paper makes use of the ICN network's in-network storage capability, deterministic latency enhanced resolution system, and name addressing capability for supporting blockchain storage optimization.

The middle two layers in boxes represent this paper work, including storage structure and collaborative replica deletion strategy. First, the storage structure includes blockchain node community division based on the enhanced resolution system and virtual chain structure that provides stored blocks indexing. Blockchain nodes within the same resolution service area are established as a partition, and a blockchain ledger is shared within the partition. The ledger is stored on the underlying network nodes in blocks to ensure that the blockchain ledger data is available in close proximity. The partition elects a supernode to maintain a virtual chain for fast indexing of blocks stored in the network. On the basis of partitioning and virtual chain, this paper proposes collaborative block replica deletion strategy, including replica number decision mechanism based on blockchain access decay relationship and replica deletion algorithm based on resource relationship.

This architecture can provide blockchain storage, blockchain index, and blockchain fetching services to various public blockchains, including Bitcoin, Ethereum, and EOS. This blockchain storage architecture makes full use of the ICN network characteristics, solves the blockchain storage scalability problem and ensures the fast access performance for blockchain users to the ledger data.

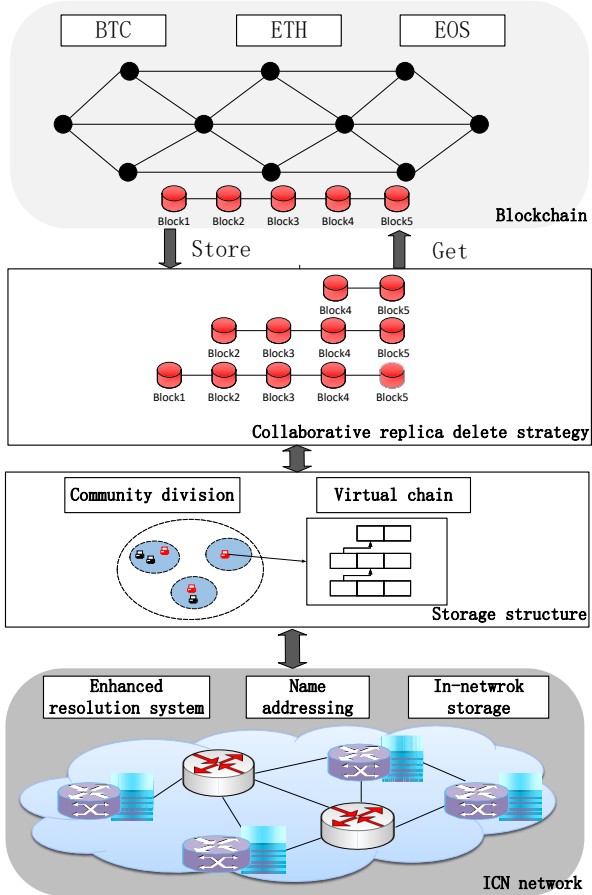

**Figure 2.** The blockchain storage architecture.

### 3.2. Blockchain Node Community Division

To ensure high data availability, scholars combine the community discovery mechanism with replica placement. The core idea is to divide nodes that are physically located close into the same partition and place replicas to ensure fast access to data. Bonvin [25] first proposes to use game theory to solve the replica placement problem. It requires time-delayed measurements to obtain complete topological information, which incurs significant network overhead, and makes it difficult to guarantee the physical location proximity. Several research teams [26] use approximation algorithms to partition the nodes into communities. The most representative one is that Liu [27] uses a distributed community discovery algorithm for community partitioning based on the request frequency and geographic location information, and adopts the greedy algorithm to place replica, which can approximate centralized replica placement algorithms. However, these community partitioning algorithms all estimate the distance between nodes by latency measuring and have high computational cost. The algorithm complexity is always more significant than $O(N^2)$, where $N$ is the amount of node data in the system.

This paper performs community division based on the enhanced resolution system, where the maximum resolution latency can be ensured by resolution nodes at each level. The theoretical analysis is shown in Figure 3. R1-1 is the resolution node in the deterministic latency enhanced resolution system, and the deterministic resolution latency of

the resolution node at this level is T ms. S1, S2, and S3 are the network devices within the resolution service area of R1-1. According to the guarantee for resolution latency, the latency T2 between S1 and the resolution node R1-1 is less than the deterministic latency T ms. Similarly, the latency T1 between S3 and R1-1 is less than T ms. Based on the above relationship, the T3 between S1 and S3 is finite, within a certain range. We can deduce that the physical locations of network devices S1 and S2 are neighboring. Using the enhanced resolution system feature, this paper divides the blockchain nodes within the same resolution service area into the same partition. It can guarantee the physical location proximity of the blockchain nodes within the partition. We use the enhanced resolution system nested relationships of service area to establish neighborhood partitions.

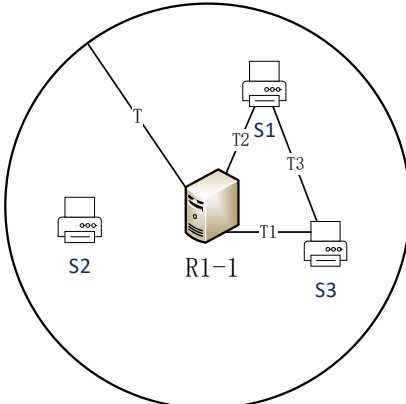

**Figure 3.** Deterministic latency characteristics in enhanced resolution system.

The work flow is as follows:

1. Blockchain application registration. Blockchain nodes register with low-level resolution nodes to establish partitions based on blockchain devices. It guarantees the physical proximity of the blockchain nodes within the partition.
2. Supernode selection and collaboration. Blockchain nodes in partition elect supernode according to the information in enhanced resolution system and node's ability. Supernodes register at high-level resolution nodes based on blockchain ledger storage services.

To describe the follow-up mechanism, we take the example of a two-tier enhanced name resolution system, including first-level resolution nodes and second-level nodes.

### 3.2.1. Blockchain Application Registration

The enhanced resolution system provides registration and resolution services for devices, services, data, and content. We use the enhanced resolution system to divide the blockchain node to partition as shown in Figure 4, where S1–S9 are blockchain nodes; R1-1, R1-2, and R1-3 are first-level resolution nodes; and R2-1 is a second-level resolution node. Blockchain nodes register with the first-level resolution node based on the blockchain device information, and we divide the blockchain nodes within the resolution service of the resolution node into a partition. Blockchain nodes in this partition can obtain other nodes' IP address according to the resolution node.

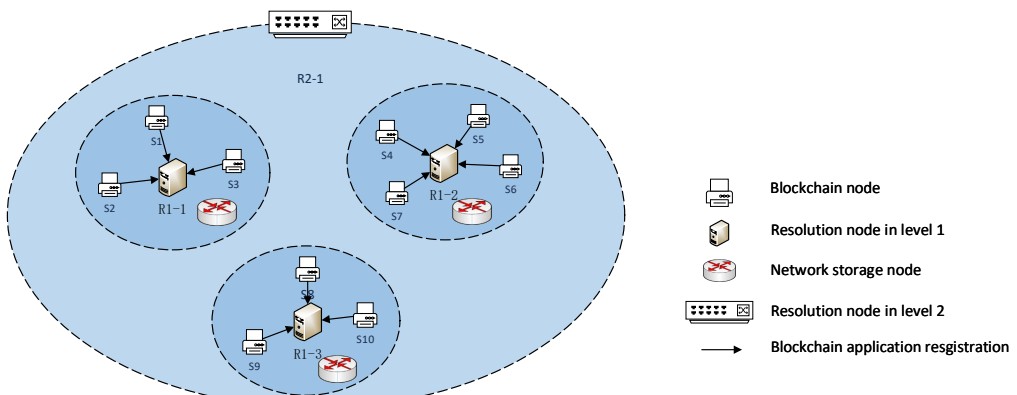

**Figure 4.** Blockchain application registration.

### 3.2.2. Supernode Selection and Collaboration

After registering the blockchain application, we divide the blockchain nodes into partitions. If each partition maintains a complete blockchain ledger in the underlying network, it is significant for the network storage load. This paper proposes blockchain ledger collaborative management between partitions to address this problem. Therefore, we design the supernode selection and collaboration as Figure 5. First, selections are conducted on blockchain nodes within the partition.

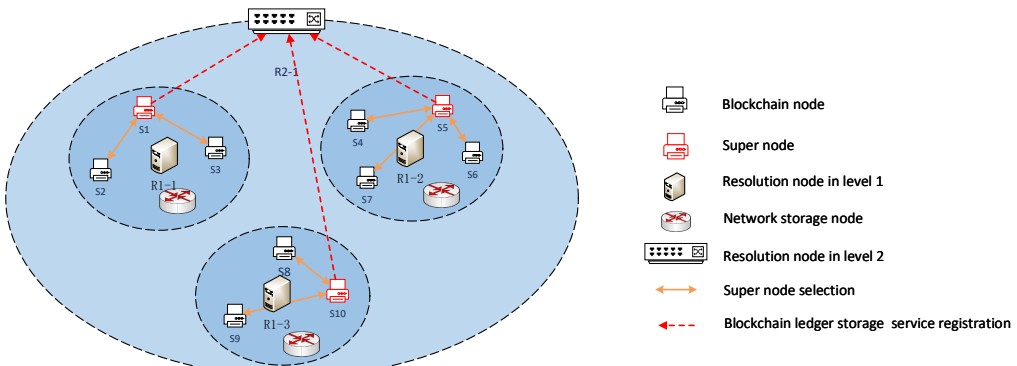

**Figure 5.** Supernode selection.

The candidate node will query in the first-level resolution node to obtain the blockchain node IP address in their partition. Then it encrypts storage space $storage_i$, block producer number $blocknum_i$, and online time $time_i$ with its private key. Supernodes maintain the virtual chain to meet the blockchain node access requirements within the partition and require a certain amount of storage space. The supernode needs to ensure trustworthiness and stability, and will not quit frequently the blockchain system. The generating block number reflects participation in the blockchain and ensures trustworthiness. The node stability is reflected based on the node online time.

Then candidate node sends the information to the other blockchain nodes. Other blockchain nodes will decrypt this information by the candidate node's public key and verify this information based on the blockchain ledger. The blockchain node will sort candidate nodes according to Formula (1) and select the best node to become the supernode in this partition, then send the voting information with its signature to the supernode, other nodes, and so on.

$$value_i = \alpha \cdot storage_i + \beta \cdot blocknum_i + \gamma \cdot time_i$$
$$s.t.\alpha + \beta + \gamma = 1 \tag{1}$$

where $\alpha$, $\beta$, and $\gamma$ are weight factors that can be dynamically adjusted according to different scenarios.

After selecting successfully, supernode will register at the second-level resolution node according to the blockchain storage service. The supernodes in other partitions are also selected, and register at the second-level resolution node. Supernodes obtain each other's IP addresses at second-level resolution nodes to build collaboration between neighboring partitions. Moreover, the maximum resolution latency guarantees that these partitions are physically located close to each other to establish nearby partitions.

Compared with existing mechanisms using delay measurement and traffic analysis, this method utilizes network characteristics to effectively reduce message interaction overhead, improve the community division accuracy, and ensure the physical location proximity of blockchain nodes within the partition.

### 3.3. Virtual Chain

The supernode in the partition stores the ledger in blocks to the underlying network node. For the fast index, we propose the virtual chain structure, which can also be used as the structural basis for block replica management in next subsection.

The virtual chain is maintained by the supernode in each partition, as shown in the Figure 6, and has the chain structure of blockchain, but does not store specific blockchain data. *Parent hash* represents the block's parent block identity hash and *Block hash* represents this block identity hash to link the virtual chain with a hash pointer. *Local address* represents the IP address information of the network node storing the block. If it is empty, it means that the block is not stored in this partition. On this basis, the virtual chain also maintains *Replica number* and *Store Node* fields, which are used to indicate the block replica number in the neighboring partition and the partition where the block replica is currently stored. It can support cross-partition block acquisition and serve as the basis for collaborative replica deletion management in the next section.

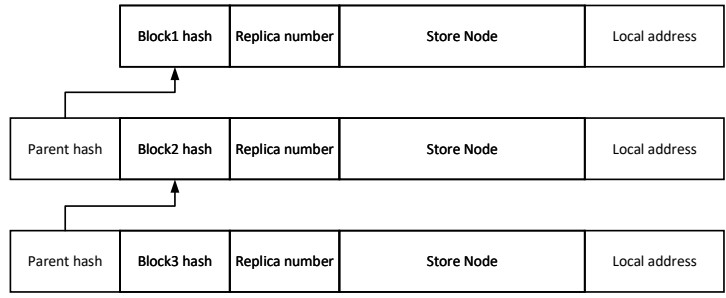

**Figure 6.** The virtual chain.

### 3.3.1. Block Storage Process

The blockchain storage architecture proposed in this paper can support the block broadcasting and consensus process, such as Bitcoin and Ethernum, and the block storage process is as follows:

(1) When a supernode receives a new block, it stores the block within the network. After storing the block file, the network returns the storage node IP address information to the supernode. The supernode maintains the address information on the virtual chain.

(2) When an ordinary blockchain node receives a new block, it will cache it locally until the block is confirmed to be on the chain and can be deleted. Meanwhile, ordinary blockchain nodes can maintain the SPV block header [2] according to their own storage

space and status, and can verify the blocks after acquisition to ensure the blockchain data untamperable.

(3) The supernodes collaborate with each other, and the virtual chain also maintains information about the block replica of neighboring partitions, including the block replica number and the partition whether the block replica is stored.

### 3.3.2. Block Acquisition Process

The block acquisition process based on this storage architecture is as follows:

(1) The user submits block acquisition request to the blockchain node, and the blockchain node will query the supernode in the partition. First, the supernode will query the local virtual chain. If the block replica is reserved in the partition, the network node IP address storing the replica will be directly returned to the blockchain node. If this partition is not available, the supernode will query to other partition supernode in step 2.

(2) When receiving request, the supernode of the neighboring partition will return the node IP address storing the block replica. After the blockchain node receives it, it will continue to obtain the target block data in step 3.

(3) Blockchain nodes obtain the target block data in the network based on the block hash and network node IP address.

### 3.4. Collaborative Replica Delete Strategy

Community partitioning based on network characteristics can guarantee proximity access to the blockchain ledger. However, each partition must pay for an entire blockchain ledger's storage and maintenance overhead. Full ledger replica storage has a high level redundancy. Existing blockchain replica management schemes use segmentation or compression to reduce the data amount and randomly delete the block data in the ledger. The result is that blockchain users cannot access some blocks locally, which impacts blockchain data availability and increases ledger access reaction time. Therefore, how to optimize the block replica management becomes the next work. We have discovered that blockchain users have an uneven accessing probability for each block in the ledger, with a typical skew feature.

To this feature, our idea is to conduct collaborative block replica deletion, as shown in Figure 7, in the neighboring partition. We propose a lightweight block replica number decision mechanism based on the decaying characteristics of blockchain ledger access. To minimize the replica deletion loss, we establish a block replica deletion model based on the load–resource relationship as a constraint. For this model, we offer a greedy block replica deletion selection algorithm.

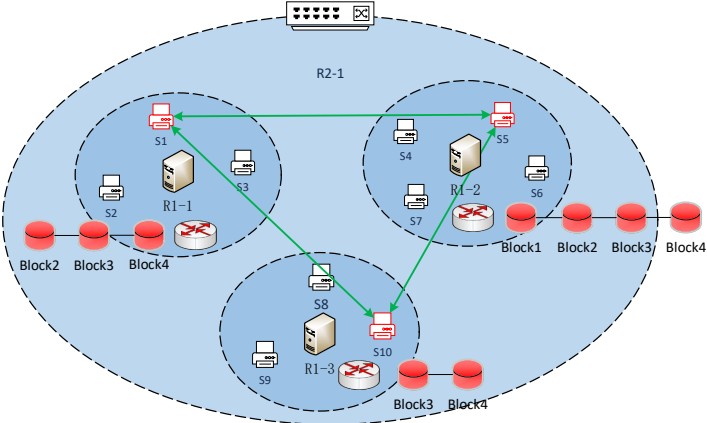

**Figure 7.** Collaborative block replica deletion.

### 3.4.1. Block Replica Number Decision Mechanism

Existing replica number decision mechanisms require statistical accesses and probabilistic transfer models, including Markov chains or gray Markov models, to forecast the accesses distribution in the next period and make replica number decisions [19,27–29]. This method is statistically expensive and ignores the unique characteristics of blockchain ledgers.

According to the latest Bitcoin data analysis paper [30], the blockchain users' access frequency to block data is correlated with the block generation time. We can see that more than 80% of user access is for blocks generated within one day, and the probability of accessing blocks with longer generation time decreases rapidly. The phenomenon is more evident in transactional blockchains, where the record content involves logistics, bills, and other financial transactions.

Analyzing the blockchain content, blockchain is essentially a ledger, which is internally linked in the form of blocks according to the temporal relationship. Each block contains a part of the ledger data. The transaction records within the block are generally contracts, stocks, notes, transaction information, and other data. Blockchain users strongly desire to access newly generated blocks and will regularly generate new transactions or blocks based on that block. As the generation time passes, the users' interest falls considerably. Blockchain users' access to the blockchain follows the network information life cycle theory. The utility value of new block data enters the peak phase quickly when it first appears, then it will undergo rapid decay with time and finally enter the decline phase.

We summarize this behavior pattern of blockchain users accessing blocks as the decay characteristics of blockchain ledger access. For modeling this decay relationship, academics have proposed many similar theories. The Ebbinghaus curve [31] models the information retention is a decaying relationship in the brain and claims that data forgetting has exponential features as Figure 8, which is then used to construct a forgetting model.

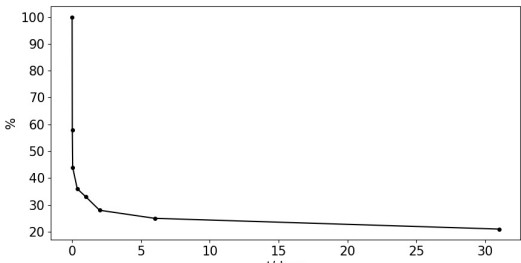

**Figure 8.** The Ebbinghaus curve.

The Ebbinghaus forgetting model curve has been widely used in recommender systems [32] and collaborative filtering [33]. The basic expressions are as following:

$$f(x) = \frac{100 \cdot k}{(\log x)^c + k} \qquad (2)$$

where $t$ denotes the time interval. $c$ and $k$ are the decay coefficient.

We find that the blockchain access interest decay follows a similar pattern, so we apply the forgetting model to the decay of blockchain access interest, using interest decay as the basis for replica number decisions.

The following is the specific implementation. It is assumed that the block files in same blockchain have similar access interest decay relationship. The block is then labeled with a decay factor when stored in the network, based on the timestamp information in the block and the forgetting model. The block decay factor fluctuates over time, showing that interest in accessing the block changes over time, which we use to determine the replica number. The calculation is as follows:

$$m_t = \begin{cases} \frac{k}{(\lg(b_{\max}.timestamp - b_t.timestamp))^c + k}, & m_t \geq m_r \\ m_r, & m_t < m_r \end{cases} \qquad (3)$$

$$R_t = m_t \cdot R_0 \tag{4}$$

where $m_t$ represents the decay factor of block t, $b_{max}.timestamp$ represents the creation timestamp of the latest block, $b_t.timestamp$ represents the creation timestamp of block $t$, $m_r$ represents the reliability requirement on the replicas number, $R_0$ represents the static replica factor, which is calculated based on the topology information, and $R_t$ represents the current replica number.

The decay factor indicates that as the block generation time passes, the interest of blockchain users in the block declines. This interest affects the block replica number, which is highest at first and then progressively drops. We can see that newly generated blocks are allocated more replicas, while old blocks are maintained with fewer replicas. Using the blockchain ledger access decay relationship for replica number decisions can greatly reduce the statistics burden in the existing replica number decision method and guarantee the user access performance to the ledger.

### 3.4.2. Replica Deletion Algorithm

After obtaining the replica number by the decay factor, how to select a deletion replica becomes a new problem. The following issues need to be considered when deleting replica:

(1) Deleting a replica directly affects the blockchain user's access performance to the ledger by adding extra access time.
(2) Each partition can withstand a certain number of requests, and inappropriate deletion of replicas concentrates the load on specific partition. As a result, these partitions become overloaded, resulting in high data locality and excessive network utilization, thus lowering system performance.

For the load resource relationship of replica deletion selection, we perform a modeling analysis to minimize the replica deletion loss, representing the user access time added by the replica deletion. The loss is expressed as follows:

$$D_j^k = \sum_i^S d_{ij}^k \cdot r_{ij}^k \tag{5}$$

where $D_j^k$ represents the total loss when deleting a block $k$ replica in partition $j$, $r_{ij}^k$ represents the access number from partition $i$ to partition $j$ for block $k$, $d_{ij}^k$ represents the loss under that access behavior, and $S$ represents the partitions number. We perform a summation analysis for the single replica deletion loss, with the following overall objective equation:

$$\sum_k \sum_j^{S_k} D_j^k \cdot x_j^k \tag{6}$$

Expand it as:

$$min \sum_k \sum_j^{S_k} \sum_i^S d_{ij}^k \cdot r_{ij}^k \cdot x_j^k \tag{7}$$

$$s.t. \sum_j^{S_k} x_j^k = num_d^k \tag{8a}$$

$$\sum_k \sum_i r_{im}^k + \sum_k \sum_i r_{ij}^k \cdot y_{jm}^k \le LC_m \tag{8b}$$

$$y_{jm}^k \le 1 - x_m^k \tag{8c}$$

$$1 \le i \le S, 1 \le j, m \le S_k \tag{8d}$$

where $x_j^k$ represents whether to delete the block $k$ replica in partition $j$, $k$'s range includes the genesis block to the newly generated block in this blockchain, $r_{ij}^k$ represents the access number to block $k$ from partition $i$ to partition $j$, $S_k$ represents the partitions number that currently stores the block $k$ replica, and $num_d^k$ indicates the deleted block $k$ replica in (8a). Equations (8b) and (8c) represent the partition load capacity constraint, $y_{jm}^k$ denotes the request to access partition $j$ is shifted to partition $m$, and $LC_m$ denotes the partition $m$ load capacity.

Based on the above block replica deletion model, we evaluate the associated models and discover that the replica deletion problem is a subset of the replica placement problem, which has been proven to be an NP-hard problem that is difficult to solve in polynomial time [34–36]. Heuristic algorithms have been used to solve the problem and approximate the optimal solution. However, this technique is typically challenging to implement, and when replicas are deleted, the same replicas will affect each other between partitions, and the effect will change dynamically.

For the replica deletion model, we approximate the optimal replica deletion distribution by deleting the replica greedily. The core idea is deleting the minor loss replica based on the replica deletion loss and the partition load state at each iteration. The specific implementation process includes message design, state calculation, and cooperative deletion algorithms.

We expand the blockchain node synchronous interaction messages to meet the collaboration requirements by adding *Delete Flag, Replica Number, State, and Regional Identifier*, as shown in Table 2. The *Delete Flag* field is used to identify the block deletion phase. The *Replica Number* field identifies the target deleted replica number. The *State* field is used to identify the loss status when partition deleting block replica. The *Regional Identifier* field is used to identify whether the partition has participated in the collaboration process.

**Table 2.** Block Replica Delete Control Message Design.

| Block Synchronization Message | *Delete Flag* | *Relica Number* | *State* | *Regional Identifier* |
|---|---|---|---|---|
| | *2 bits* | *2 bytes* | *1 bytes* | *2 bytes* |

This algorithm selects the delete partition based on the partition states and uses the greed algorithm idea to delete the replica with the worst states. The state calculation includes the deletion replica loss and load condition.

First is the replica deletion loss calculation, which is the increase in access time when a replica is deleted and requires consideration of access number, access distribution, distance received by the partition supernode, and the partition load condition.

$$
\begin{aligned}
d_{ij}^k &= \begin{cases}
\min_{m \in S_k \wedge m \notin D_{k'}} \{dis(i,m)\} \cdot e_k / bw, i = j \\
\min_{m \in S_k \wedge m \notin D_{k'}} \{dis(i,m) - dis(i,j)\} \cdot e_k / bw, i \neq j
\end{cases} \\
D_{k'} &= \left\{ D | D \subseteq S_k \wedge |D| = num_d^k \right\} \\
D_j^k &= \sum_i^S d_{ij}^k \cdot r_{ij}^k \\
1 &\leq i \leq S, 1 \leq j, m \leq S_k
\end{aligned}
\tag{9}
$$

where $D_j^k$ represents the loss of partition j deleting block $k$ replica, $d_{ij}^k$ represents the behavior of partition i requesting block k from partition j, $k$'s range includes the genesis block to the newly generated block in this blockchain, $D_{k'}$ represents the partition set of deleted block $k$ replica, $S_k$ represents the partition where the block $k$ replica is stored, $bw$ represents bandwidth, and $e_k$ represents the block k data amount.

$$LC_j = \sum_k \sum_i r_{ij}^k, 1 \leq j \leq S_k, 1 \leq i \leq S \tag{10}$$

where $LC_j$ represents the load limit of partition j supernode.

$$S_j = \beta \cdot D_j^k + (1 - \beta)/LC_j \tag{11}$$

The above two are then weighted and summed to indicate the partition current state, which is used as the basis for replica deletion.

The collaborative block replica deletion algorithm consists of three phases: the deletion message construction phase, the partition deletion state interaction, and the block replica deletion and update. SN represents the supernode.

The first phase builds the delete message phase and initiates the collaborative deletion process. As shown in Algorithm 1, the target block k replica number is first calculated based on the decay factor in Formula (4) and then compared with the current block *k* replica number in the local virtual chain. If the target number is less than the current number, the block replica deletion phase is entered. In the message, set the *Deletion Flag* field to 1 and the *Replica Number* field to the target replica number, and fill in the *State* field with its own partition status value according to Formula (11). Initialize all partition *Regional Identifiers* fields containing block *k* replica to 0 and set its own *Regional Identifier* field to 1, representing that the partition has participated in the block replica deleting. After the message is constructed, the supernode forwards it to other supernodes that have stored block *k* replicas that are not involved in the collaboration.

---

**Algorithm 1** Deletion message construction phase.

---

1: **for** block $k$ in blockchain **do**
2:      The SN retrieval block $k$'s replica number in virtual chain
3:      Compute the block $k$ target replica number according to Formula (11)
4:      **if** block $k$ target replica number is smaller than current replica number **then**
5:          Set *Replica Number* to block $k$ target replica number,set *Delete Flag* to 1, set *State* Field to SN's state
6:          Set *Regional Identifier* for current SN to 1, other Regional Identifier to 0
7:          Forward the packet to other SN contains block $k$ replica
8:      **end if**
9: **end for**

---

The second phase, comparing the states of the individual partition, is used to assess each partition, deleting replica loss. As shown in Algorithm 2, after receiving the message, other supernodes calculate the block replica number according to the Formula (4), and if the target replica number is the same, they enter the deletion process. Set the *Regional Identifier* field representing itself to 1. It also calculates its state value according to the Formula (11) and compares it with the *State* field in the message. If the status value is less than the *State* field in the message, the *State* field updates. Keep forwarding messages until the *Regional Identifiers* of all replicas of storage block $k$ are set to 1 and the second phase is completed.

---

**Algorithm 2** Block deletion state interaction.

---

1:  The SN receives a deletion message
2:  **if** some *Regional Identifier* is 0 **then**
3:      The SN retrieval block *k*'s replica number in virtual chain
4:      Compute the block *k* target replica number according to Formula (11)
5:      **if** block *k* target replica number is equal to *Replica Number* **then**
6:          **if** The current SN state is smaller than *State* Field **then**
7:              Update the *State* field to current SN state
8:          **end if**
9:          Set *Regional Identifier* for SN contains block *k* replica to 1
10:     **end if**
11:     Forward the packet to other SN contains block *k* replica
12: **end if**

---

The third phase consists of block replica deletion state updates. According to Algorithms 3 and 4, the supernode updates its own virtual chain after deleting replica, while setting the Deletion Flag field to 2, then setting its own the Regional Identifier field to 1 and the other Regional Identifier field to 0. Forward this message to all other supernodes that store block k replica. Other supernodes receive this message and update virtual chain.

---

**Algorithm 3** Block replica deletion and update.

---

1:  The SN receives partition deletion state interaction message
2:  **if** SN state is equal to *State* field **then**
3:      SN delete the block *k* replica and update virtual chain for block *k*
4:      Set the *Delete Flag* to 2
5:      Set *Region Flag* to 1, other *Region Flag* to 0
6:      Forward the packet to all SN contains block *k* replica
7:  **end if**

---

**Algorithm 4** Block replica deletion and update response.

---

1:  The SN receives a block replica deletion and update message
2:  Update virtual chain for block *k*

---

## 4. Evaluation and Performance Analysis

In this section, we will evaluate the proposed blockchain storage architecture. The experimental platform is Intel Xeon(R) CPU E5-26090@2.40 GHz, 16 GB memory and 2 TB hard disk, and the operating system is Centos 7.9. This experiment's blockchain data originates from the Xblock website [37], an open-source website that offers a variety of public blockchain data for research. Our simulation platform uses OMNet++ [38] an open-source network simulation framework and Blocksim [39], an open-source blockchain network simulation framework. The experiment consists of three parts, decay factor analysis, blockchain storage architecture availability evaluation, and replica management strategy comparison.

### 4.1. Decay Factors Analysis

This experiment is used to evaluate the impact of different decay factors on data acquisition, using Bitcoin transaction data provided by the xblock website and the access distribution. The simulation platform is Blocksim, and the partition node number is 10, 50, and 100, while it performs 100-time data accesses under each decay factor set of k and c (Table 3). Here we assume a static factor $R_0$ for the topological nodes number. The analysis metrics are the average data acquisition time, the replica data compression rate, and the request hit ratio.

**Table 3.** Parameters in experiment 1.

| Parameter (s) | Setting (s) |
| --- | --- |
| Simulation platform | Blocksim |
| Partition node number | 10, 50, 100 |
| Experiment run time for each scenario | 100 |

First of all, we can see the trend of the access block time from the Figures 9–11. Each curve's block access time increases significantly as the c-value rises. However, it can be seen that $k = 4$ has a relatively flat growth trend, while $k = 3$ and $k = 5$ have a sharp jump in growth in the first period and a rapid increase in the later period against the growth of the c-value, and the three tend to be close to each other in the latest period. For replica data volume changes, the replica data volume all drops quickly as the c-value rises. With the shift in c-value, The drop in data volume is more pronounced for $k = 3$, while $k = 4$ and $k = 5$ are relatively smoother.

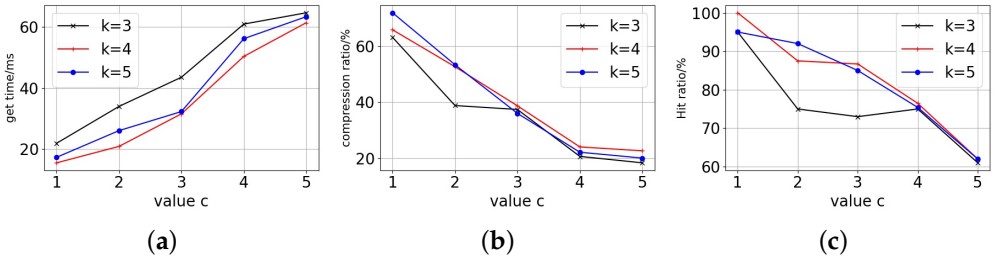

**Figure 9.** Particion node = 10. (**a**) Get time. (**b**) Replica data compression. (**c**) Hit ratio.

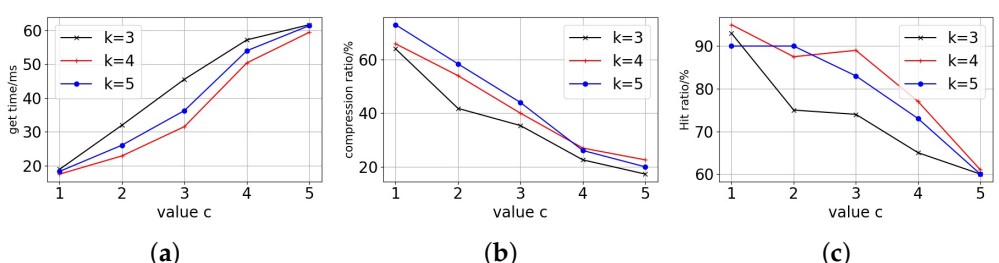

**Figure 10.** Particion node = 50. (**a**) Get time. (**b**) Replica data compression. (**c**) Hit ratio.

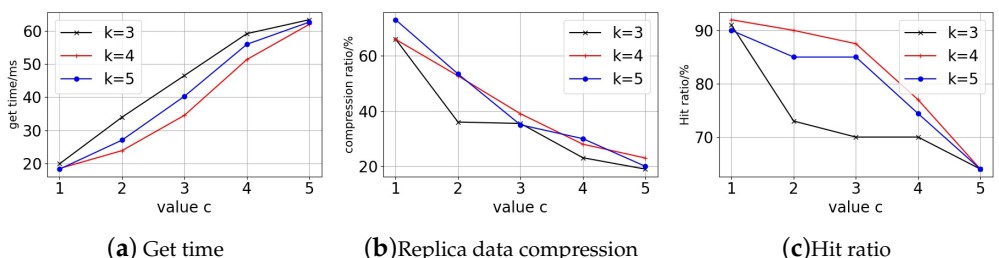

(**a**) Get time     (**b**)Replica data compression     (**c**)Hit ratio

**Figure 11.** Particion node = 100. (**a**) Get time. (**b**) Replica data compression. (**c**) Hit ratio.

Finally, in terms of the replica access hit ratio, it can be seen that both $k = 4$ and $k = 5$ have significantly better hit ratio than that of $k = 3$ in the early period and become closer in the later period, and this trend is also reflected in the data acquisition time. The fundamental cause is that when the c-value is higher, the replica number of all curves is low. Excessive replica deletion lowers hit ratio and require data fetching and accessing across partitions, resulting in longer response time for users accessing the ledger. A smaller c-value with more replicas means that blockchain users can access data that are close to them, but replica storage costs are higher. For stability and access performance, we choose

$k = 4$ and $c = 3$ as the decay factor. The decay factors can guarantee access performance and save the copy storage overhead by more than 60%.

### 4.2. Blockchain Storage Architecture Comparison

This experiment evaluates the data availability in different blockchain storage architectures.Comparison architectures include BC-store [22] and KARAKASA [18]. BC-store is an IPFS-based blockchain storage architecture that stores blockchain ledgers in the IPFS network. KARAKASA is a blockchain storage architecture that organizes blockchain nodes into DHT clusters. Each blockchain node is responsible for maintaining a portion of the blockchain ledger data to reduce the storage pressure. To make KARAKASA more comparative, we build Chord and Kroode DHT protocols in KARAKASA architecture. In the experiment, BC-store represents the BC-store storage architecture. KARAKASA-chord and KARAKASA-kroode represent the KARAKASA storage architecture, and NB-RD illustrates the architecture proposed in this paper.

The analysis indicator is data acquisition time. We conduct comparative availability analysis for blockchain storage architectures with node scales of 1000, 3000, and 5000. Experiment parameters are shown in the following Table 4. KBR protocol is semi-recursive. It means that the initial node encapsulates the message and sends it to the nearest node in the routing table. The message is recursively forwarded until it reaches the target node, and the response message is sent directly to the starting point. The overlay protocol is IEEE 802.11 with RTS/CTS Extension. The node number is stable without churn.

**Table 4.** Parameters in experiment 2.

| Parameter (s) | Setting (s) |
| --- | --- |
| Simulation platform | OMNet++ |
| KBR protocols | Semi-recursive |
| Overlay protocol | IEEE 802.11 with RTS/CTS extension |
| Churn generator types | NoChurn |
| Topology size | 1000, 3000, 5000 |
| Packet size | 512 Byte |
| Communication link delay | 50 ms |
| Network bandwidth | 20 Mbps |
| Simulation time | 200 s |

As can be seen from the violin distribution of acquisition time in Figures 12–14, the storage architecture proposed in this paper is obviously superior to other schemes. When the node scale is 1000, the mean data acquisition time in BC-store is 0.4831 s, with a variance of 0.0971. The mean data acquisition times in KARAKASA architecture are 0.2162 s and 0.5733 s, respectively, with variances of 0.0124 and 0.0452. The mean data acquisition time of the architecture in this paper is 0.1099 s, with a variance of 0.0103 s. When the node scale is 3000, the mean data acquisition time in BC-store is 0.6584 s, with a variance of 0.0956. The mean data acquisition times in KARAKASA architecture are 0.2904 s and 0.6053 s, with variances of 0.0241 s and 0.0394, respectively. The mean data acquisition time of the architecture in this paper is 0.1096 s, a variance of 0.0066. When the node scale is 5000, the mean data acquisition time in BC-store is 0.8109 s, with a variance of 0.1511. The mean data acquisition times in KARAKASA architecture are 0.4697 s and 0.7606 s, with variances of 0.0526 s and 0.0433, respectively. The mean data acquisition time of the architecture in this paper is 0.1296 s, with variance of 0.0659. It can be seen that the data acquisition time in the BC-store increases rapidly with the increase of node scale. The acquisition time in KARAKASA also grows slightly when node scale grows, albeit at a slower rate than BC-store, whereas NB-RD grows slowly and remains steady as node scale grows.

This is because BC-store uses an IPFS network, and IPFS uses the kademlia algorithm, a distributed hash table for indexing. Furthermore, the IPFS network lacks an active replica dispersion strategy, resulting in low availability. The cluster scheme based on DHT is based

on greed algorithm during partition, which can reduce the remote data acquisition situation. Therefore, as the blockchain nodes increase, the DHT-based scheme's data acquisition time is significantly better than BC-store. The NB-RD proposed in this paper is significantly better than the DHT schemes in data acquisition time. The reason is that the blockchain nodes in the same partition are all within the deterministic delay range, ensuring the physical location proximity and short transmission distance during data acquisition. However, the DHT scheme adopts a logical partition, and the nodes' actual distance will affect the acquisition time, and NB-RD can be dynamically adjusted based on the number of blockchain nodes in a partition. Therefore, with the blockchain node's scale increasing, the data acquisition time does not increase significantly and is highly scalable.

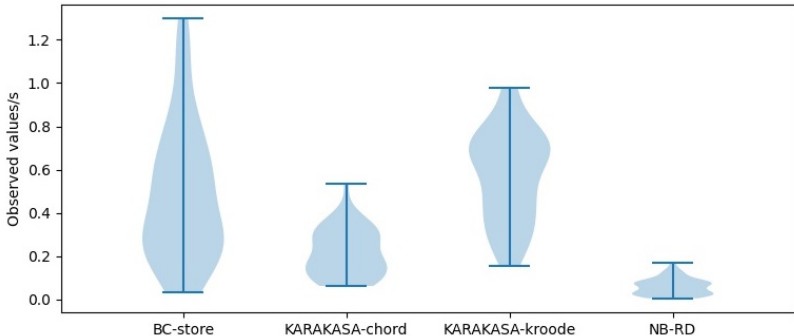

**Figure 12.** Availability performance in node = 1000.

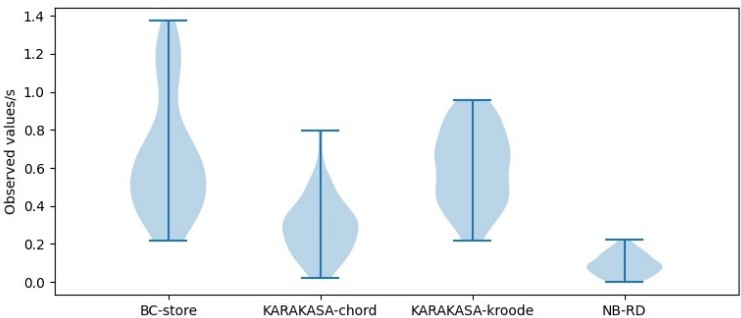

**Figure 13.** Availability performance in node = 3000.

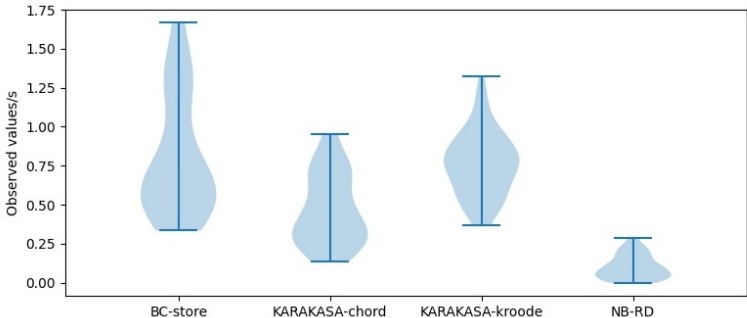

**Figure 14.** Availability performance in node = 5000.

The traditional blockchain storage architecture, as seen in [22], uses a full replica storage architecture, which has apparent advantages during data access. The data acquisition time of Bitcoin nodes is 0.0192 ms. However, compared with this NB-RD, the data acquisition time is about 100 ms, and the storage pressure of a single blockchain node is significantly reduced.

### 4.3. Replica Management Strategy Comparison

This experiment is used to evaluate different replica management strategies. As a comparison, there is a full replica strategy and a segmentation strategy [28,29], where the full replica strategy stores an entire blockchain ledger in each partition, whereas the segmentation strategy slices the blockchain ledger and then deletes the segments using the MCMC random algorithm. Each node only needs to keep a portion of the ledger data. In the experimental evaluation, we use *NB-fullreplica* to denote the full replica policy, *NB-segmented* to indicate the segmentation policy, and *NB-RD* to denote the replica deletion management policy proposed in this paper.

On the Blocksim platform, the blockchain node number is 100, the static factor $R_0$ is 10, and the analyzed metrics are data acquisition time, access hit ratio, block replica number, and replica data amount (Table 5). The blockchain utilizes the Bitcoin transaction data and access provided by the xblock website.

**Table 5.** Parameters in experiment 3.

| Parameter (s) | Setting (s) |
|---|---|
| Simulation platform | Blocksim |
| Static factor $R_0$ | 10 |
| Value k | 4 |
| Value c | 3 |
| Topology size | 100 |
| Simulation time | 10,000 s, 20,000 s |

Figure 15 shows that the NB-RD is close to the full replica strategy in acquisition time, with the average acquisition time being just 5.72% percent longer than the full replica strategy, which is substantially better than the segmentation strategy. Meanwhile, NB-RD has a replica hit rate of 95.2%, which is higher than the segmentation strategy's 86% and has a better hit ratio. This method has significantly smaller replica number than the full replica strategy. The amount of data preserved is only 57.2% of the whole replica strategy, which is better than the existing segmentation strategy. Of course there is a spike in fetch time for NB-RD, mainly because burst accesses to old blocks need to be fetched from other partitions after replica deletion, resulting in a steep increase in fetch time. However, fewer spikes occur compared to the segmentation strategy because this method exploits the decaying relationship of blockchain accesses, where new blocks are assigned more replicas, and old blocks are appropriately reduced in the replica number. This replica management method is more in line with the laws of blockchain access, which greatly improves the replica access hit ratio and reduces the replica data volume while ensuring blockchain access performance.

From the above Figure 16, we can see that the NB-RD continues to perform well as the simulation time grows and the block data increases. First, in data acquisition time, NB-RD is close to the full replica strategy. Although with more blocks generated, the average data acquisition time increased slightly by 7.74% compared to the full replica strategy. However, the access hit ratio of NB-RD is 89%, compared with 63% under the segmentation strategy, which is much better than the segmentation strategy in terms of access performance. NB-RD is more stable, whereas the segmentation strategy's replica hit rate drops dramatically as the number of blocks increases, resulting in numerous "spikes" in data fetching time and a significant decline in access performance. Meanwhile, the number of replicas maintained of NB-RD is much smaller than the full replica strategy, only 40.4% of the data volume of the full replica strategy, which is better than the segmentation strategy. However, we can see that the NB-RD "spike" increases access time since there will be occasional access to old blocks as the number of blocks increases. The replica number of old blocks shrinks after deletion, making it difficult to meet local requirements and needs to access across partitions, which is time consuming.

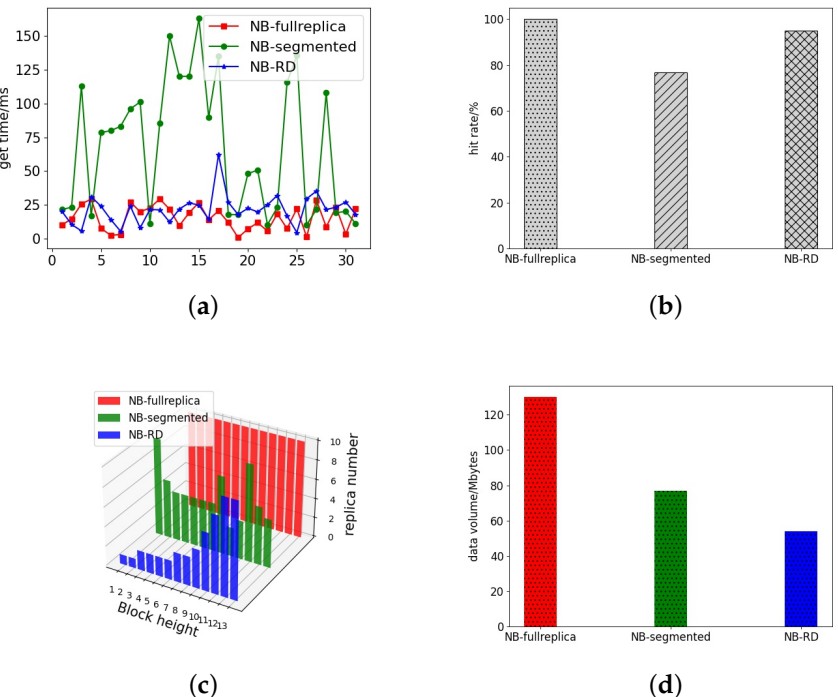

**Figure 15.** Simulation time = 10,000 s. (**a**) Data acquisition time. (**b**) Access hit rate. (**c**) Block replica number. (**d**) Replica data amount.

However, compared to the segmentation strategy, the NB-RD has a minor increase in access time because it considers deletion loss and node load during replica deletion to ensure efficient block access across partitions and avoid long-distance block data acquisition and queuing issues caused by high node load.

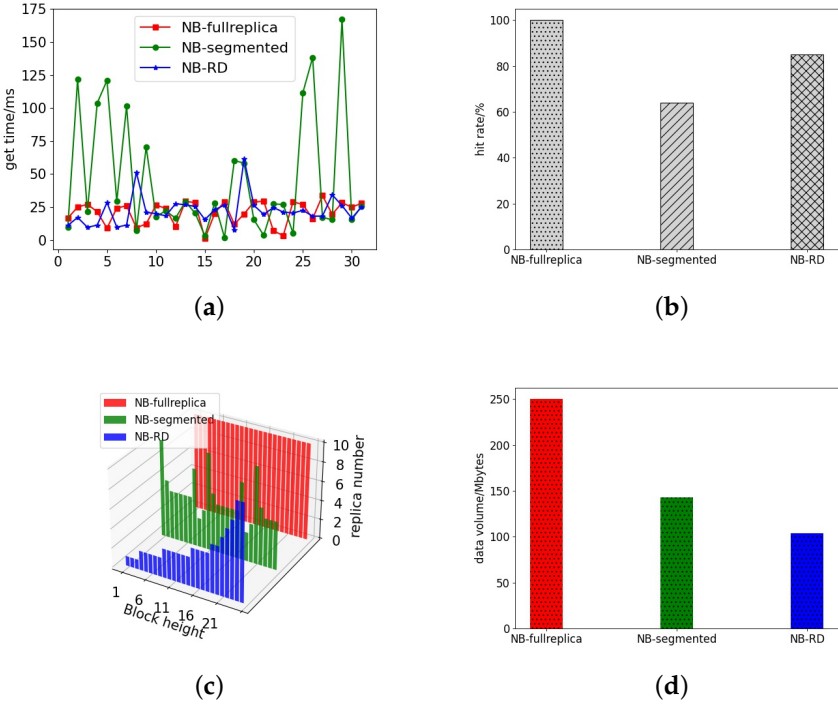

**Figure 16.** Simulation time = 20,000 s. (**a**) Data acquisition time. (**b**) Access hit rate. (**c**) Block replica number. (**d**) Replica data amount.

## 5. Conclusions and Future Work

Due to its distinctive features, blockchain technology has attracted much attention and has been actively developed. However, along with the rapid development, the blockchain scalability problem is becoming more and more prominent, which has seriously restricted the blockchain's potential to advance. For the blockchain scalability problem, the existing research solutions suffer from a high average response time for users to access the ledger. In this paper, we propose a blockchain storage architecture based on the ICN network by using a deterministic time-delay enhanced resolution system for community division and a virtual chain fast index to guarantee the proximity and quick access of blockchain nodes to access the ledger. Based on blockchain node community division, for the existing blockchain replica management scheme with poor access problems, collaborative block replica deletion between nearby communities is carried out. It uses blockchain ledger access decay characteristics to decide the replica number and conducts replica deletion based on deletion loss and load, drastically reducing the storage cost while maintaining the performance of blockchain nodes accessing the ledger. It has been experimentally proven that this architecture has a lower average response time and better stability for users to access blockchain ledgers compared to IPFS-based and DHT-based blockchain storage architectures. After experimental comparison and analysis, the replica data amount is reduced by 57.2% compared to the full replica policy.The blockchain node access performance decreases by only 5.2% compared to the full replica policy, which ensures the blockchain users access performance.

In the architecture proposed in this paper, blockchain nodes store blockchain ledger data in the network. When blockchain users request transaction data in the ledger, if the transaction information provided by users is missing, all block files need to be traversed during retrieval. It will bring high transmission costs and a poor user experience. The next step is to determine further the block based on transaction characteristics. It can be used in conjunction with this study to improve the average user's blockchain service response time.

**Author Contributions:** Conceptualization, H.F. and J.W.; methodology, H.F. and J.W.; software and data curation, H.F.; validation and writing—original draft preparation, H.F.; writing—review and editing, H.F., Y.L. and J.W.; supervision, J.W. and Y.L.; project administration, J.W.; funding acquisition, J.W. All authors have read and agreed to the published version of the manuscript.

**Funding:** This work was funded by the Strategic Leadership Project of the Chinese Academy of Sciences: SEANET Technology Standardization Research System Development (Project No. XDC02070100).

**Acknowledgments:** We would like to express our gratitude to Jinlin Wang and Yang Li for their meaningful support for this work.

**Conflicts of Interest:** The authors declare no conflict of interest.

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
