# Peer review of "A Blockchain Storage Architecture Based on Information-Centric Networking"

_electronics, doi:10.3390/electronics11172661_

Round 1
Reviewer 1 Report
This paper proposed a blockchain storage structure based on the information-centric networking. It's very interesting and the structure of this manuscript is organized well. However, the writing of this manuscript required professional editing before publishing. There are some comment for authors here.
1. Title of this manuscript should use full name instead of ICN. I think "information-centric networking" should appear in the title instead just appear in the abstract or keywords.
2. The full manuscript require professional editing service for English checking, grammar as well as format of citations. The authors should check "guideline for author" from Electronics as reference.
3. Abstract is clear, but when reading to Section 1 Introduction, too many citations disrupt logic of story telling. Just paragraph One along, there are 13 references here. Readers may not have ability to know these 13 papers' relationship to this first paragraph of Section 1 Introduction, especially "4) other fields[10][11][12][13]." The authors should list clearly these 4 papers' contributions instead of just say "other field".
4. From Line 58 to Line 78, there are too many acronyms required to be defined before using. The authors should also put related reference in this area because these acronyms should be from other people's research.
5. The author should also provide research scope, research assumption, research limitation in the introduction section.
6. Citation format are difference between "Line 108 to Line 125" and "Line 126 to Line 159". The authors should check "guideline for authors" and make sure consistency.
7. "Line 172 and Line 173" with 4 references do not provide clear meaning to support this research. I do not know whether these 4 models compared with the authors model are good or bad or other relationships.
8. All variables in this whole manuscript should be italic.
9. Line 304: "Where" -> "where" and without indent.
10. Line 359: "step 2" is fuzzy in this paragraph. There is no clear Step 1 immediately before and Step 3 immediately after this "step 2".
11. 5 lines below Line 430: "Where" -> "where" and without indent.
12. Equation (6) and Equation (7) should define range of k.
13. Equation (8) should define range of i, j, k, m.
14. Line 434: there is NO constraint 8.1, constraints 8.2 and 8.3 in this manuscript.
15. caption format of Figure 9 required adjustment.
16. Line 459: "where" should be no indent.
17. Line 462 Equation (10) should define range of i, k.
18. Algorithm 1 uses "block k" all the way to Algorithm 2, 3 and 4. But before Algorithm 1, it's called "partition k".
19. The authors should check "guideline for author" from Electronics for reference list format.
Reviewer 2 Report
Topic of the manuscript is not so close to the fields of my expertise. Accordingly, my review can be and is based only on the general insight into the manuscript and nothing else.
The manuscript is well written. It contains very good insight to the research in the considered field, to the results reached till now, as well as a respectable comparative analysis. However, in current form, the manuscript contains several technical flaws.
Firstly, figures given in the manuscript must be technically improved. Although figures are not so complex, the introduced notations are so small that they are almost invisible in the most figures (see, for example, Figs. 1, 2, 8, 9, 10-17). This figures must be improved. In addition, references used in the manuscript are actual, but should be substantially improved. Namely, the manuscript includes a lot of references (59 ones). Moreover, the number of cited conference papers is especially high (almost third part of the included references are conference papers). I suggest reference reduction through the reduction of conference papers.
Round 2
Reviewer 2 Report
Authors have fulfilled almost all my requirements from the first round of review process. Only legends related to Figures 9-11 should be enlarged. As well, English language should be checked once more (by native speaker, if possible).
Note once more that topic of the manuscript is not so close to the fields of my expertise. Accordingly, my review can be and is based only on the general insight into the manuscript and nothing else.
